# Drying Enhances Signal Intensities for Global GC–MS Metabolomics

**DOI:** 10.3390/metabo9040068

**Published:** 2019-04-05

**Authors:** Manuel Liebeke, Erik Puskás

**Affiliations:** Max-Planck Institute for Marine Microbiology, Celsiusstr. 1, 28359 Bremen, Germany; htfmov@gmail.com

**Keywords:** metabolomics, sample preparation, gas chromatography, derivatization, silylation

## Abstract

We report here that a straightforward change of the standard derivatization procedure for GC–MS metabolomics is leading to a strong increase in metabolite signal intensity. Drying samples between methoxymation and trimethylsilylation significantly increased signals by two- to tenfold in extracts of yeast cells, plant and animal tissue, and human urine. This easy step reduces the cost of sample material and the need for expensive new hardware.

## 1. Introduction

Metabolomics, the measurement of hundreds of metabolites at once, has changed our capability to describe and understand cellular and organismal physiology. To fully analyze the diverse chemical space of metabolites, like small amino acids and complex lipids in samples ranging from body fluids to cells or whole tissue, complementary technologies are needed. To this end, combinations of nuclear magnetic resonance spectroscopy, liquid-chromatography (LC) or gas chromatography (GC) coupled to a mass-spectrometer (MS) have been used [1,2,3,4]. GC–MS based metabolomics covers a wide range of primary metabolites which are difficult to analyze with other methods in a single run [5,6,7] and employs derivatization chemistry to increase the number of compounds in the gaseous phase. Moreover, GC–MS is broadly available and has high separation power, robust quantitation, high selectivity, and sensitivity. A major challenge in metabolomics is the metabolite identification, for GC–MS there are standardized mass-spectral databases for thousands of metabolites (e.g., Fiehn-DB [5], GOLM-DB [8], NIST Mass Spectral Library, Wiley Registry) which represents an advantage for metabolite assignment compared to other methods. Due to the technical properties and its wide coverage of metabolites from central metabolic pathways to steroids and xenobiotic molecules, GC–MS metabolomics is an essential tool for the investigation of biological processes. Technical development is ongoing, with recent advances enabling the generation of ever more complex metabolic profiles containing 1000’s of deconvoluted peaks per sample using the newest mass-spectrometry detectors [9,10]. Hardware advances aside, sample preparation approaches differ widely across sample types, as large divergence across different laboratories. In contrast, derivatization procedures have fundamentally remained the same over the last decade or two. Derivatization for comprehensive GC–MS metabolomics requires the transformation of polar functional groups, like –OH, -NH_2_, –COOH, –SH, to gain more volatile products. Routinely, this is achieved with trimethylsilylation (Trimethylsilylated-, TMS-versions of the compound). Alternatives to silylation are acylation/esterification derivatization protocols [11], yet these methods do not detect certain compound classes, such as sugars, for a comprehensive metabolomics approach [12].

In this study, we aimed to improve the sensitivity of the most commonly used two-step derivatization (1st methoxymation, 2nd trimetylsilylation) method to enable every laboratory with existing GC–MS equipment to improve metabolome wide signal intensity. Here we introduce a protocol for derivatization by changing only the derivatization procedure and yielding much higher or more metabolites signals, or both, without cutbacks in reproducibility and robustness.

## 2. Results

The simplest step to increase metabolite signal height is to reduce the volume of derivatization reagents and therefore increase the concentration of compounds in the sample. Our derivatization procedure was optimized using a representative set of primary metabolites with varying physical and chemical properties. For this purpose, we included amino acids, organic acids, sugars, a fatty acid, a nucleotide base, phosphorylated sugars and a sterol into a synthetic equimolar mixture (see Appendix A). We kept temperatures for derivatization low (37 °C) to prevent metabolite breakdown and used commonly accepted standard incubation times (1st step is 90 min methoximation followed by 2nd step 30 min trimethylsilylation with N,O-bistrimethyl-silyltrifluoro-acetamide (BSTFA) [5]. Our initial effort for reducing the final volume of the derivatization liquid was to leave out the methoximation step to see the potential maximum signal height occurring only from trimethylsilylation. GC–MS data for the synthetic mixture of metabolites showed signals for amino acids and organic acids, but with much lower intensities compared to the conventional 2-step protocol (data not shown). As described in previous reports, sugars gave rise to multiple peaks and strongly decreased signal intensities [13]. Next, we tested the reduction of the sample volume by evaporating pyridine, the main solvent from the methoximation reaction (1st step), using a gentle N_2_ gas stream. Following trimethylsilylation of the dried residue (see Figure 1a), we observed a significant signal increase compared to the standard 2-step procedure (see Figure 1b,c, and Appendix A). Amino acids and organic acids increased on average by 80.9% and 49.6%, respectively. Sugars exhibited an average increase of 40.4% (see Figure 1c). Only a few peaks showed a significant decrease in intensities (see Figure 1b,d), those peaks were mainly related to the derivatization reagents or side-products thereof [14] (see Appendix A). In the metabolite mixture, only glycine (3-TMS) and lysine (4-TMS) peaks showed a signal decrease using the drying method. For both metabolites a strong shift of the derivatization reaction towards a different TMS derivative occurred, abundant glycine (2-TMS), and lysine (3-TMS) peaks were detected using the drying method (see Figure 1d and Appendix A). The drying method also increased signals of metabolites, like phosphorylated sugars or phosphorylated acids, which have been reported to show degradation during GC–MS metabolomics experiments (see Appendix A). Reproducibility of the drying method was comparable across all metabolites tested, whereas some metabolite peaks showed a slightly higher coefficient of variation (CV) compared to the standard method (mean CV_standard_ = 7.7% vs. mean CV_drying_ = 9.4%, including glycine and lysine peaks with very high CVs due to the derivative shift). Reducing the final sample volume from 180 µL to 100 µL (1.8-fold reduction) increased the compound concentration in the sample and led to an average 2.7-fold signal intensity gain. Clearly, this signal enhancement could not solely be explained by pyridine evaporation after derivatization step one (see Figure 1a). This observation was further surprising as the general assumption is that pyridine acts as an acid scavenger during the derivatization and will catalyze the trimethylsilylation reaction [15]. In contrast, our data imply that metabolite derivatization does not benefit from pyridine addition. We further tested the drying method for different metabolite concentrations and showed comparable linear signal responses to the standard method (see Appendix A).

Based on our results using a synthetic metabolite mixture, we applied the drying method using methoxymation and trimethylsilylation with BSTFA to real-world complex metabolome extracts from plants, yeast cells, animal tissue, and urine. All of the tested sample types showed significantly increased peaks for the vast majority of compounds (see Figure 1e, Appendix A). Yeast extract, the most complex sample in terms of number of peaks, showed the highest boost in signal considering the proportion of increased number and integral of peaks (83.1%, 4408 features of 5302 changed more than 1.5-fold), followed by animal tissue (78.7%), urine (67.3%), and plant extract samples (58.1%), respectively (see Figure 1e, Appendix A). A considerably smaller proportion of signals decreased in all tested samples, matching observations from the synthetic metabolite mixture (e.g., derivatization artifacts, specific derivative shifts like described above for some amino acids). Surprisingly, the increased signal intensity in the complex metabolome samples exceeded the effect observed for the synthetic metabolite mixture, e.g., alanine increased in standard mixture 2- fold and 8-fold in urine samples, (see Figure 1f, Appendix A). The increased metabolite complexity of real-world samples allowed us to observe the effect of the drying step on more metabolite classes such as aromatic acids (e.g., benzoic acid in plant metabolome, 2-fold increase), phosphorylated metabolites (e.g., 3-phosphoglycerate in yeast metabolome, 3-fold increase) (see Figure 1f). In urine, amino acids (e.g., alanine, 8-fold; leucine 4-fold) and bases (e.g., thymine, 4-fold) increased above average signal increase, compared to the standard method (see Figure 1f and Appendix A). Overall, our global metabolomics method, presented here, increased signals by 2.4-fold across all metabolite signals and samples, with some metabolites showing very high increases in intensity. One example is the metabolite hippuric acid found in urine (see Figure 1f, Appendix A), which has been shown to be a biomarker of many physiological or pathological conditions, or both, with a strong association to diet and the intestinal microbiota [16]. Using our method, the detection was considerably improved (39-fold increase for the 2-TMS derivative; 2-fold for the 1-TMS derivative). The signal boost for urine is opening the possibilities to design highly sensitive metabolite assay methods for diverse medical applications.

The enhanced signal in the presence of a complex sample background, like the hundreds of different metabolites in cell extracts, led us to test the possibility that drying favors derivatization reaction kinetics of individual metabolites. We prepared a mixture of metabolites and added complexity by increasing the number of different metabolites in the sample. Our experiment confirmed the findings from the complex metabolite extract samples. That is, adding further metabolites to a sample increased the signal for individual metabolites (e.g., glycine and succinic acid), whereas others were not affected (e.g., sugars like sucrose and ribitol) (see Appendix A). Such signal increase correlated with sample complexity was reported previously, but a mechanistic explanation has not been formulated yet [17].

Next, we checked if other derivatization reagents used for GC–MS metabolomics also result in a signal gain by drying. Firstly, we tested the reagent MTBSTFA (N-tert-Butyldimethylsilyl-N-methyltrifluoroacetamide) which produces tert-butyl dimethylsilyl derivatives. A commonly used derivatization procedure for the synthetic metabolite mixture was applied, only marginal signal increases with the drying step involved were observed compared to the normal procedure (median signal fold change = 1.2) (see Appendix A). We further tested the trimethylsilylation reagent MSTFA (N-methyl-N-trimethyl-silyltrifluoro-acetamide), here the strongest increase of metabolite signals was achieved (median signal fold change = 3.1) (see Appendix A), surpassing even the above-described signal boost by chemical closely related BSTFA.

## 3. Discussion

The observed signal gain across derivatization reagents and the different sample types demonstrates the substantial analytical and economic value of our method. Importantly, the method can easily be implemented at negligible cost in any lab using GC–MS based metabolomics. To facilitate the implementation of the drying method, we provide the model files for 3D printing of a drying device which was designed in this study and costs only a fraction of commercially available lab equipment (see Appendix A). The presented drying device allows for a batch processing of 24 samples at once and together with the here presented GC–MS method all these samples are analyzed within one day. We presented the benefits of evaporating the methoximation reagent for untargeted metabolomics GC–MS analysis for a variety of sample types. On these grounds, we anticipate that other methods using similar derivatization chemistry, e.g., pesticide or steroid analysis, will also benefit. Overall, we believe that our approach could have a substantial impact in the field of derivatization-dependent GC–MS analysis in general and on global metabolomics in particular.

## 4. Materials and Methods

### 4.1. Chemicals

All chemicals were purchased from Sigma-Aldrich (Germany) in high purity grade (LC–MS grade). BSTFA and MSTFA were purchased from CS Chromatographie-Service (Germany). Derivatization related reagents were all anhydrous and handled with care to prevent the introduction of moisture.

### 4.2. Metabolite Standard Mixture

We selected representative compounds from different classes of metabolites for this study (see Appendix A). Stock solutions of standards were prepared in double-distilled water at a final concentration of 20 mM, except the stock solutions of myristate, palmitate, and cholesterol which were prepared in ethanol. The mixtures of standards were dried under vacuum using a vacuum rotation evaporator (Extractor Plus^®^, Agilent, settings: 30 °C, 4 h, 1000 rpm, VAQ).

### 4.3. Metabolite Extraction

Human urine from volunteers was prepared using an established urine metabolomics protocol [18]. Urine was aliquoted into 200 µL samples and prepared, including the depletion of urea with urease, precipitation of urease with methanol and extraction of metabolites. The plant metabolome extract was generated from seagrass leaves (*Posidonia oceanica*, collected in the Mediterranean sea, the island of Elba) and was prepared using the method for global plant metabolomics [6]. Briefly, shock-frozen leaf tissue was ground with a mortar and pestle (all cooled with liquid N_2_), 100 mg powder was extracted with methanol and strong agitation, chloroform and water were added and the sample mixed. After phase-separation, the upper layer was taken for metabolite analysis. Lysogeny broth (Sigma Aldrich) was prepared after the manufacturer’s recipe, excluding the addition of salts, and 100 µL was used as a sample for the yeast cell metabolite extract. Animal tissue (common earthworm, *Lumbricus terrestris*) was collected and extracted as described previously [19]. Briefly, the shock-frozen whole animal was ground with a mortar and pestle (all liquid N_2_ cooled), 50 mg powder was extracted as described above for the plant tissue. All final extracts were dried under vacuum (Extractor Plus^®^, Agilent; settings: 30 °C, 2 h, 1000 rpm, VAQ).

### 4.4. Derivatization Procedures

For step 1, the oximation, 80 µL methoxylamine hydrochloride (anhydrous) in pyridine (0.005% H_2_O) (20 mg ml^−1^) was added to the dry sample and incubated under shaking (1350 rpm) at 37 °C for 90 min. Afterward, the sample was centrifuged for 30 s with a table-top centrifuge to spin down any remaining liquid from the lid. The sample was subsequently evaporated to dryness under a constant stream of nitrogen for 60 min at 21 °C and subsequently trimethylsilylated with 100 µL BSTFA for 30 min, 1350 rpm, 37 °C. Samples not treated with a drying step were left at 21 °C for 60 min before trimethylsilylation to have comparable controls. The MSTFA experiments were done in m the same way, exchanging BSTFA against MSTFA only. For the derivatization with MTBSTFA we used 20 µL methoxymation reagent (1st step, 90 min at 37 °C) and in the 2nd step 80 µL N-(tert-butyldimethylsilyl)-N-methyltrifluoroacetamide (60 min at 70 °C).

### 4.5. GC–MS Method

The analysis was performed on an Agilent 7890B gas chromatograph connected to an Agilent 5977A MSD. Samples were run according to methods recommended for metabolome databases [5]. Briefly, 1 µL sample was injected in splitless mode with an Agilent 7693 autosampler injector into deactivated splitless liner (Agilent 5190-3167, splitless, single taper, glass wool). Separation was done on an Agilent 30 m DB5-MS column with a 10 m DuraGuard column. Oven temperatures were set to 60 °C for 2 min, and then headed with 10 °C min^−1^ up to 325 °C and set to hold for 5 min at 325 °C. Metabolites were assigned using the NIST 2.0 Library and verified by injection of pure compounds under the same chromatographic conditions.

### 4.6. GC–MS Data Analysis

Pairwise comparisons between sample groups from different treatments were done with Chemstation AIA exported files using XCMS online (https://xcmsonline.scripps.edu/) [20] set to default parameter for file import (GC/Single Quad –centWave). Datasets are available under metabolights [21] https://www.ebi.ac.uk/metabolights/ entry MTBLS807. The mean fold change (log2) between the conventional method and the drying method was calculated using the minimum noise level for zero feature intensities to avoid fold change with infinity results.

## Figures and Tables

**Figure 1 metabolites-09-00068-f001:**
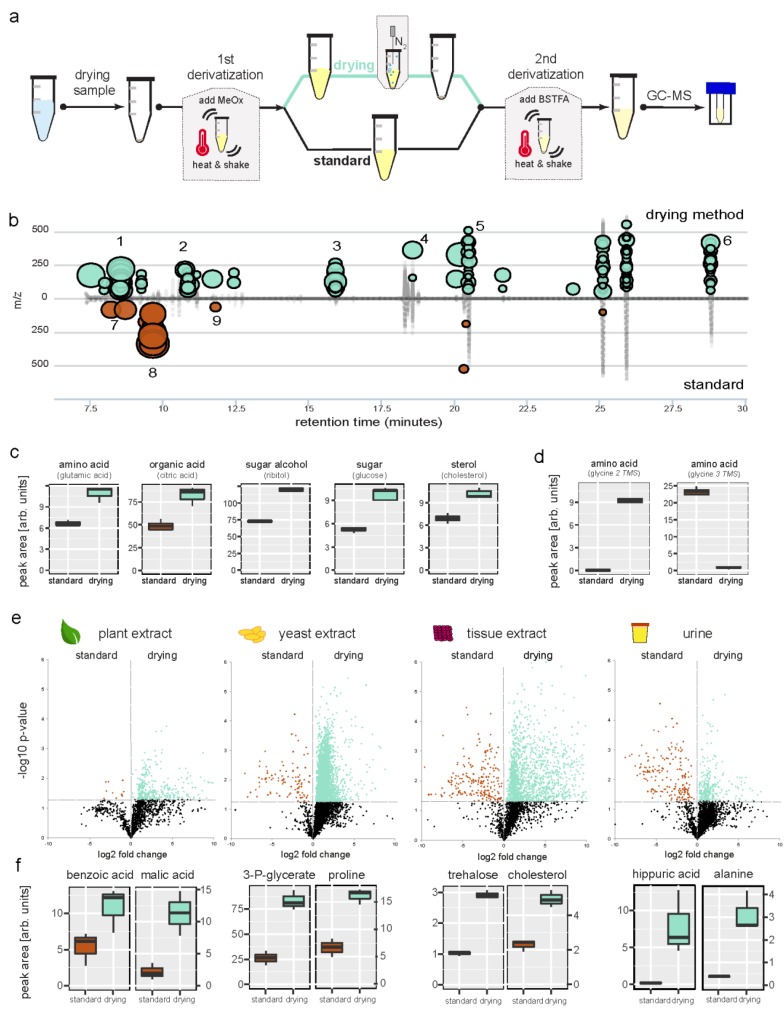
Enhanced metabolite signals by implementing a drying step into global metabolomics GC–MS derivatization. (**a**) Drying derivatization method in comparison with the standard sample preparation scheme. (**b**) Synthetic metabolite mixture showing increased peaks using the drying method against the standard method visualized via chromatographic cloud plot (XCMS). ‘Bubbles’ indicate significantly changed features between tested methods (>1.5 fold change; *p* < 0.05, Welch *t*-test). Increased metabolites (green) are alanine,1; valine, 2; glutamic acid, 3; citric acid, 4; glucose, 5; cholesterol, 6. Decreased peaks (red) are derivatization side products, 7 and 8; glycine 3TMS, 9 (see Appendix A for identifications). (**c**) Enhanced signal area for metabolites of synthetic metabolite mixture using the drying method. (**d**) Derivatization (trimethylsilylation) product is changing for glycine using the drying method. (**e**) Metabolome wide signal enhancement in plant-, yeast cell-, tissue, and urine samples. Comparing fold changes of all detected GC–MS features, standard method against the drying method. Each sample type was analyzed in triplicates using both methods. Significantly increased features with drying method are highlighted in green, decreased features in red (*p* value < 0.05, Welch *t*-test). Mean fold change across all signals is 2.2 for plant extracts; 2.3 for yeast cells; 3.6 for tissue; and 1.6 for the urine sample. (**f**) Metabolites with large signal increase using the drying method, from plant, yeast, tissue and urine samples.

## Data Availability

Datasets are available under metabolights [21] https://www.ebi.ac.uk/metabolights/ MTBLS807.

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
