# Peer review of "Drying Enhances Signal Intensities for Global GC–MS Metabolomics"

_metabolites, 2019, doi:10.3390/metabo9040068_

Round 1

Reviewer 1 Report

The manuscript by Liebeke and Puskas presents a method for alteration of the derivatisation process to improve the signal of metabolite detection by GC-MS.  This communication is of great value to the metabolomics community especially in situations where sample is limiting i.e. low abundance cell systems, clinical blood spots etc.

The communication presents results from experiments that are well designed, utilising both authentic standards and also biological matrices.  The results give a compelling argument as to why GC-MS analysts should consider adopting this methodology to their research.

I think that this communication should be published, but I do have a few comments that I would like to have addressed by the authors.

Comments:

1) The results presented cover a wide range of metabolites from various classes.  All the metabolites chosen though are in my opinion generally very robust metabolites and are not prone to degradation.  I would like to see data presented from more labile metabolites such as pyruvate, glutamine and a variety of phosphorylated metabolites other than 3PGA.  Phosphorylated metabolites that I would like included are: G6P, F6P, R5P, Ru5P and 6PG to name a few.  All these metabolites are prone to degradation, but are some of the most biologically relevant metabolites - so are of immense interest.

2) I would like the authors to comment on how this method would perform for experiments where the sample numbers are >50. The metabolomics community strives to improve analytical reproducibility in order to ensure that the biological variability of interest can be observed - it's not clear in this communication that this type of batch derivatisation would be as reproducible as wanted.

3) This follows on from comment 2.  A large proportion of the community would use robot autosamplers such as CTC-PAL or Gerstel to perform online automated in time derivatisation of samples to overcome issues of reproducibility.  How would the authors perceive their methodology fitting in with these automated workflows.

General comments:

- The english in this manuscript is a bit clumsy at times and can make it a bit hard to read.

- As an example - line 32-24, the sentence doesn't make sense

- There is something strange happening with reference numbering.  Some are in square brackets, while some aren't.

Overall this communication of value to the community.  It is definitely novel and some of the observations will be very thought provoking to GC-MS experts in metabolomics.  I think that it should be published once the comments above have been addressed.

Reviewer 2 Report

The paper is well written. The scientific rigour is very high and many experiments have been carried out.

I can suggest some little modifications.

1.      Abstract: “We report here that” is not catchy sentence. I suggest to improve the section with sentences more impersonal.

2.      Introduction:

a.      Lines 25-28, Page 1: Sentence too long and complex! The authors make it clearer.

3.      In general, the first time that we cite a chemical abbreviation, it is better to indicate the whole name: in particular

a.      Line 54, page 2: BSTFA

b.      Line 141, page 4: MSTFA

4.      In some point of the main text, references are not reported in square brackets (lines 39-40 pag1; lines 168, 195 pag 5). Control all references in the text, please.

5.      Material and methods. If it is possible, for “Metabolites” guidelines, the authors underline the

subparagraphs  (with bold or italic).

a.      In the subparagraph “Metabolite extraction”, the sentence “with methanol and strong agitation, chloroform and water was added and the sample mixed” is repeat twice. I suggest to make it clearer and to avoid to repeat.

6.      Author contribution: at the last, delete semicolon.
